# Evaluation of the Impact of Climate Change on the Water Balance of the Mixteco River Basin with the SWAT Model

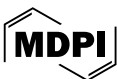

Gerardo Colín-García [1] , Enrique Palacios-Vélez [2] , Adolfo López-Pérez [2],* , Martín Alejandro Bolaños-González [2] , Héctor Flores-Magdaleno [2] , Roberto Ascencio-Hernández [2] and Enrique Inoscencio Canales-Islas [3]

1   National Institute for Forest, Agriculture and Livestock Research (INIFAP), Center of Chiapas Experimental Field of the South Pacific Regional Research Center, Ocozocoautla–Cintalapa International Highway Km 3.0, Ocozocoautla de Espinosa 29140, Mexico; colin.gerardo@inifap.gob.mx

2   Hydrosciences, Postgraduate College, Campus Montecillo, México-Texcoco Highway Km 36.5, Montecillo 56264, Mexico; epalacio@colpos.mx (E.P.-V.); bolanos@colpos.mx (M.A.B.-G.); mhector@colpos.mx (H.F.-M.); ascenciohr@colpos.mx (R.A.-H.)

3   National Institute for Forest, Agriculture and Livestock Research (INIFAP), Santiago Ixcuintla Experimental Field of the Central Pacific Regional Research Center, Mexico–Nogales International Highway Km 6.0, Santiago Ixcuintla 63300, Mexico; canales.enrique@inifap.gob.mx

\*   Correspondence: adolfholp@colpos.mx; Tel.: +52-595-110-9896

**Abstract:** Assessing the impact of climate change is essential for developing water resource management plans, especially in areas facing severe issues regarding ecosystem service degradation. This study assessed the effects of climate change on the hydrological balance using the SWAT (Soil and Water Assessment Tool) hydrological model in the Mixteco River Basin (MRB), Oaxaca, Mexico. Temperature and precipitation were predicted with the projections of global climate models (GCMs) from the Coupled Model Intercomparison Project Phase 6 (CMIP6); the bias was corrected using CMhyd software, and then the best performing GCM was selected for use in the SWAT model. According to the GCM MPI-ESM1-2-LR, precipitation might decrease by between 83.71 mm and 225.83 mm, while temperature might increase by between 2.57 °C and 4.77 °C, causing a greater atmospheric evaporation demand that might modify the hydrological balance of the MRB. Water yield might decrease by 47.40% and 61.01% under the climate scenarios SP245 and SSP585, respectively. Therefore, adaptation and mitigation measures are needed to offset the adverse impact of climate change in the MRB.

**Keywords:** bias correction; climate scenarios; management plans; water yield; water resources

## 1. Introduction

The United Nations Framework Convention on Climate Change defines climate change (CC) as "a change of climate which is attributed directly or indirectly to human activity that alters the composition of the global atmosphere and which is in addition to natural climate variability observed over comparable time periods" [1]. This is frequently expressed as a variation in precipitation and temperature [2], and it refers to a change in climate compared to the average conditions of the atmosphere over a period of time, resulting from the direct or indirect alteration of its composition [3]. Its most important drivers are greenhouse gas (GHG) and aerosol emissions, as well as changes in the albedo of the Earth's surface, which cause an energy imbalance [4].

Anthropogenic activities are currently considered the main drivers of climate change, leading to climatic catastrophes with irregular patterns [5]. Despite this knowledge, human activities continue to increase GHG emissions, making climate change a growing global issue [6]. This is to such an extent that, according to recent assessments by Richardson [7], the atmospheric carbon dioxide ($CO_2$) concentration threshold that ensures the climate stability of the planet (350 ppm and radiative forcing of 1 W m$^{-2}$) was significantly exceeded, as 417 ppm had been reached in 2022 (radiative forcing of 2.91 W m$^{-2}$).

Understanding the impact of climate change is achieved with future climate projections, generated with different Global Climate Models (GCMs) which are currently grouped into the global climate modeling framework of the Coupled Model Intercomparison Project Phase 6 (CMIP6), an international project to compare the results of climate model simulations carried out according to a common protocol [8]. GCMs use scenarios called Shared Socioeconomic Paths (SSPs) in their latest generation to make climate change projections. For example, SSP245 represents a medium-term socioeconomic development scenario with an average radiative forcing of 4.5 W m$^{-2}$, and SSP585 considers intensive development driven by the use of fossil fuels, with a high radiative forcing of 8.5 W m$^{-2}$, both for the year 2100 [9].

The connection between climate change and water resources is central to social prosperity, since this phenomenon causes alterations in the hydrological cycle that affect the availability of water resources in terms of both quantity and quality [10]. Some analyses of global precipitation data have concluded that 10% of the world's most intense rainfall has already increased by about 80% and will continue rising at a faster rate; on the other hand, there are parallel increases in the risk of occurrence of droughts, leading to the increasing occurrence and severity of flooding, landslides, and to scarcity of water resources [11]. Regarding runoff volumes, a marked reduction is expected from the different CMIP climate scenarios, causing a decrease in water availability in various parts of the world [12,13].

The rate of increase in annual mean temperature in Mexico is considerably higher than the global average rate. For example, the increase in the mean air temperature in Mexico since the beginning of the 20th century was 1.69 °C (1.59–1.8 °C), while the rise in the global mean temperature was 1.23 °C [14]. Additionally, Mexico's geographic location and relief directly influence the susceptibility to hydrometeorological events such as droughts, floods, and landslides [15], which increase its exposure to climate change, projecting a decrease in precipitation and a temperature rise [16]. Today, the mean annual precipitation is 740.00 mm, mainly from June to September. However, due to decreasing precipitation, deforestation, and soil degradation, 90% of its territory has recently been affected by droughts, altering hydrological processes and affecting agricultural and livestock production systems [17].

The Mixteco River Basin (MRB) is located at the place of origin of the Balsas Hydrological Region. Its functional dynamics are highly distorted due to altered hydrological processes, river soil degradation, water stress, and potential diffuse pollution [18]. The Mixteca region is characterized mainly by its degree of environmental degradation, deforestation level, water scarcity, and accelerated desertification processes as a result of anthropogenic activities [19,20], as well as the increasing frequency of droughts, being an area with a temperate and semi-arid climate [21]. Additionally, according to the National Atlas of Vulnerability to Climate Change (ANVV, in Spanish) produced by the National Institute of Ecology and Climate Change (INECC, in Spanish), 65% of the MRB municipalities currently have high or very high levels of vulnerability to climate change [22].

The relationship between water resources and the hydrological cycle is crucial for climate change, as the latter will directly affect water availability and the spatial and temporal variations in runoff, mainly due to changes in significant climate factors (precipitation and temperature) [2,23,24]. This will impact current practices in designing and managing natural resources [25].

Hydrological modeling is a tool that allows for analysis of the behavior of the hydrological processes that occur within a basin [26]. Therefore, it is widely used in multiple research studies for the comprehensive management of water resources [27] and for evaluating the impact of climate change [28]. In recent decades, the Soil and Water Assessment Tool (SWAT) model has been used to quantify the hydrological response to different management practices and climate change scenarios in different environmental conditions and basin sizes [29,30]. The SWAT model is a physically based semi-distributed hydrological model that allows for the calculation of the water balance components at spatial and temporal levels and production of runoff and sediments in basins [31]. Mainly, it is used to calculate the availability of water in a basin [32].

Assessing the impact of climate change on hydrological processes at spatial and temporal levels is essential to define comprehensive management of water resources in a basin [33]. Therefore, the main objective of this study is the analysis of the impact of climate change on the hydrological balance components of the Mixteco River Basin (MRB), located in the northeast of Oaxaca state, Mexico, based on short-, medium-, and long-term projections of precipitation and temperature from several Global Climate Models (GCMs) under the SPP245 and SSP585 climate scenarios, applying hydrological modeling with the SWAT model.

## 2. Materials and Methods

### 2.1. Study Area

The study area corresponds to the Mixteco River Basin (MRB). Its main channel originates from the confluence of the Tlaxiaco and Juxtlahuaca rivers; its delimitation begins from the mountainous region to the Mariscala hydrometric station (17°51′45″ N; 98°08′58″ W), with an approximate length of 180.16 km, and it becomes a major tributary of the Balsas River [34] (Figure 1).

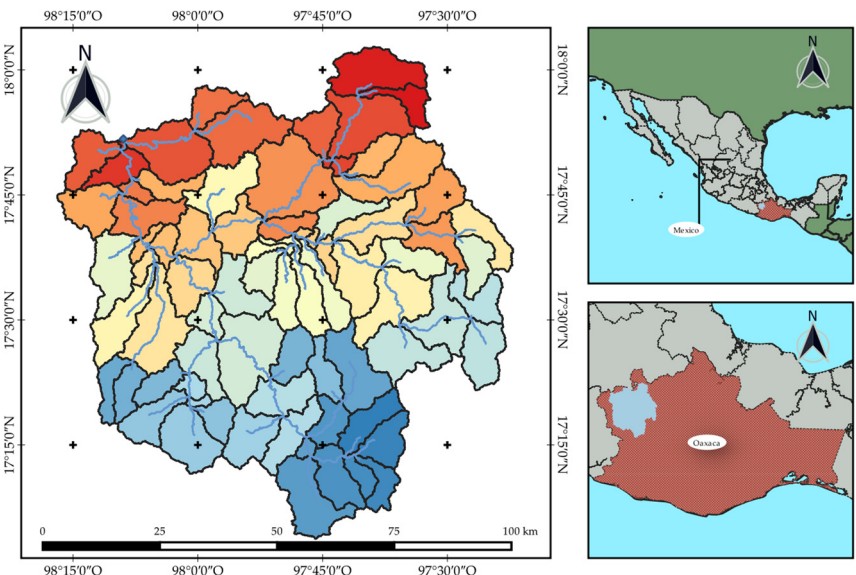

**Figure 1.** Location of the Mixteco River Basin, Oaxaca, Mexico.

The MRB comprises 6559.20 km$^2$ over an elevation range of 1.040 m to 3.366 m, with an average of 2.213 m (Figure 2a). Most of the basin has a steep relief, with an average slope of 34.29% (Figure 2b). Additionally, its rainfall regime ranges from May to October, with a mean annual precipitation of 733.55 mm. The monthly temperature varies between 16.94 °C and 22.09 °C; the lowest and highest temperatures occur in January and May, respectively.

The plant cover of the MRB corresponds mainly to forests (44.95% of the basin; Figure 2c). The most representative plant cover is oak forest (BENC, 21.54%), followed by pine-oak forest (ENPI, 10.74%), pine forest (PINE, 6.43%), oak–pine forest (ENPI, 3.31%), tascate (BTAS, 2.26%), and mountain cloud forest (BOMM, 0.67%). The area also harbors induced pastures (PASI, 21.85%), rainfed agriculture (TEMP, 19.02%), and low deciduous forest (SEBC, 10.06%). The remaining area is covered by irrigation agriculture (RIEG, 1.77%), scrubland (MATO, 1.58%), induced palm-tree forest (PALM, 0.44%), medium-density urban area (URMD, 0.24%), low-density urban area (URLD, 0.07%), and water bodies (WATER, 0.04%). The main soil classes in the MRB, according to the FAO WRB Classification System (FAO, 2014), are Leptosols (LP; 48.25%), followed by Cambisols (CM; 35.80%), Regosols (RG; 8.00%), Fluvisols (FL; 7.16%), and Vertisols (VR; 0.79%) (Figure 2d).

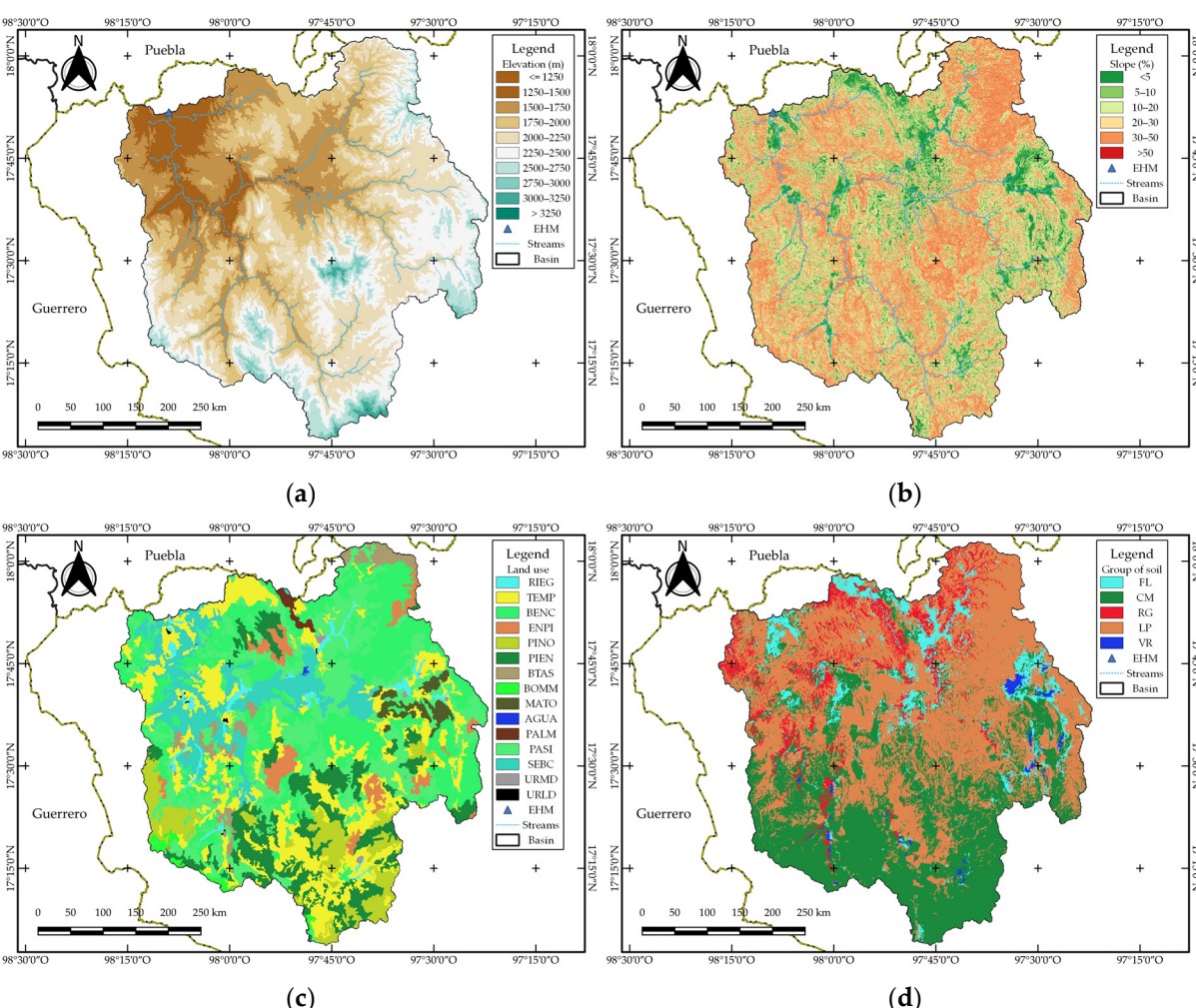

**Figure 2.** Characteristics of the MRB: (**a**) elevation range; (**b**) slope range; (**c**) plant cover; (**d**) soil type.

## 2.2. SWAT Hydrological Model

The SWAT (Soil and Water Assessment Tool) program is a semi-distributed hydrological model that simulates the different components of the hydrological balance in continuous time (annual, monthly, and daily) at spatial and temporal levels. Its primary function is to estimate runoff and sediment production and predict the impact of soil management practices on water quality in large basins with different elevations, plant cover, and soil types [31,35]. It simulates the terrestrial phase of the hydrological cycle based on hydrological balance (Equation (1)):

$$SW_t = SW_0 + \sum_{i=1}^{n} \left( R_{day} - Q_{surf} - ET - W_{seep} - R_f \right) \tag{1}$$

where $SW_t$ is the final soil water content of day $n$ (mm); $SW_0$ is the initial soil water content of day $i$ (mm); $t$ is the simulation period (days); $R_{day}$ is the total precipitation of day $i$ (mm); $Q_{surf}$ is the surface runoff of day $i$ (mm); $ET$ is the evapotranspiration of day $i$ (mm); $W_{seep}$ is the water leaching through the soil profile of day $i$ (mm), and $R_f$ is the return flow of day $i$ (mm).

Surface runoff refers to the amount of precipitation not lost through interception, infiltration, and evapotranspiration, which occurs when the precipitation rate is greater than the infiltration rate. For its calculation, the SWAT model uses the Soil Conservation Service curve number method (SCS) [36] and the Green and Ampt method [37], however, the latter requires precipitation data in sub-daily or hourly time intervals that are difficult

to obtain, which frequently makes its application difficult. Therefore, the SCS curve number method based on Equation (2) was used in this study.

$$Q_{surf} = \frac{\left(R_{day} - 0.2S\right)^2}{R_{day} + 0.8S} \tag{2}$$

where $Q_{surf}$ is the accumulated surface runoff (mm); $R_{day}$ is the total amount of precipitation for a day (mm); $S$ is the retention parameter (mm), which varies spatially due to changes in the characteristics of the land surface (soil type, land use, slope, and management practices) and, in addition, could be temporarily affected due to changes in soil water content. The retention parameter is estimated with Equation (3).

$$S = 25.4 \left(\frac{1000}{CN} - 10\right) \tag{3}$$

where $CN$ is the curve number for a day, and its value is determined by land use management practices, soil type permeability, and the soil's hydrologic group. The SWAT model utilizes the classification of the United States Natural Resources Conservation Service (NRCS), which groups soils into four hydrologic groups (A, B, C, and D) based on their infiltration rates being high, moderate, slow, and very slow, respectively.

The SWAT model calculates soil evaporation and plant transpiration separately, based on available meteorological data. Potential evapotranspiration (PET) can be calculated using the Penman–Monteith (PM), Priestley–Taylor (PT), and Hargreaves–Samani (HS) methods. The PM method requires variables such as solar radiation, air temperature, wind speed, and relative humidity; meanwhile, the PT method only needs solar radiation, air temperature, and relative humidity. The HS method solely employs air temperature. Therefore, based on the available information, this method was utilized for calculating the PET of the MRB.

Figure 3 illustrates the climate change assessment process using the SWAT model and its integration with Global Climate Models (GCMs), according to Saade et al. [38]. First, the river basin is delimited from the Digital Elevations Model (DEM), then, it is subdivided into multiple sub-basins and each of these is divided into Hydrological Response Units (HRU), depending on the slope, soil type, and plant cover. The simulation of the hydrological cycle can be divided into terrestrial phase and routing: the first determines the water load in the main channel, and the second is the flow of these loads through the different channels of sub-basins toward the outputs [39]. The simulation is performed for each HRU, including precipitation, interception, surface runoff, evapotranspiration, leaching, lateral flow through the soil profile, and return flow of shallow aquifers; this provides greater certainty and better describes the hydrological balance of the basin [40].

Furthermore, it employs a unique plant growth model for any vegetation type, differentiating between annual and perennial plants and estimating the amount of water and nutrients extracted from the root zone, transpiration, and biomass production/yield [41].

Table 1 shows the input data for the SWAT hydrological model, including the digital elevation model, spatial distribution of plant cover and soil type, and meteorological and hydrometric data.

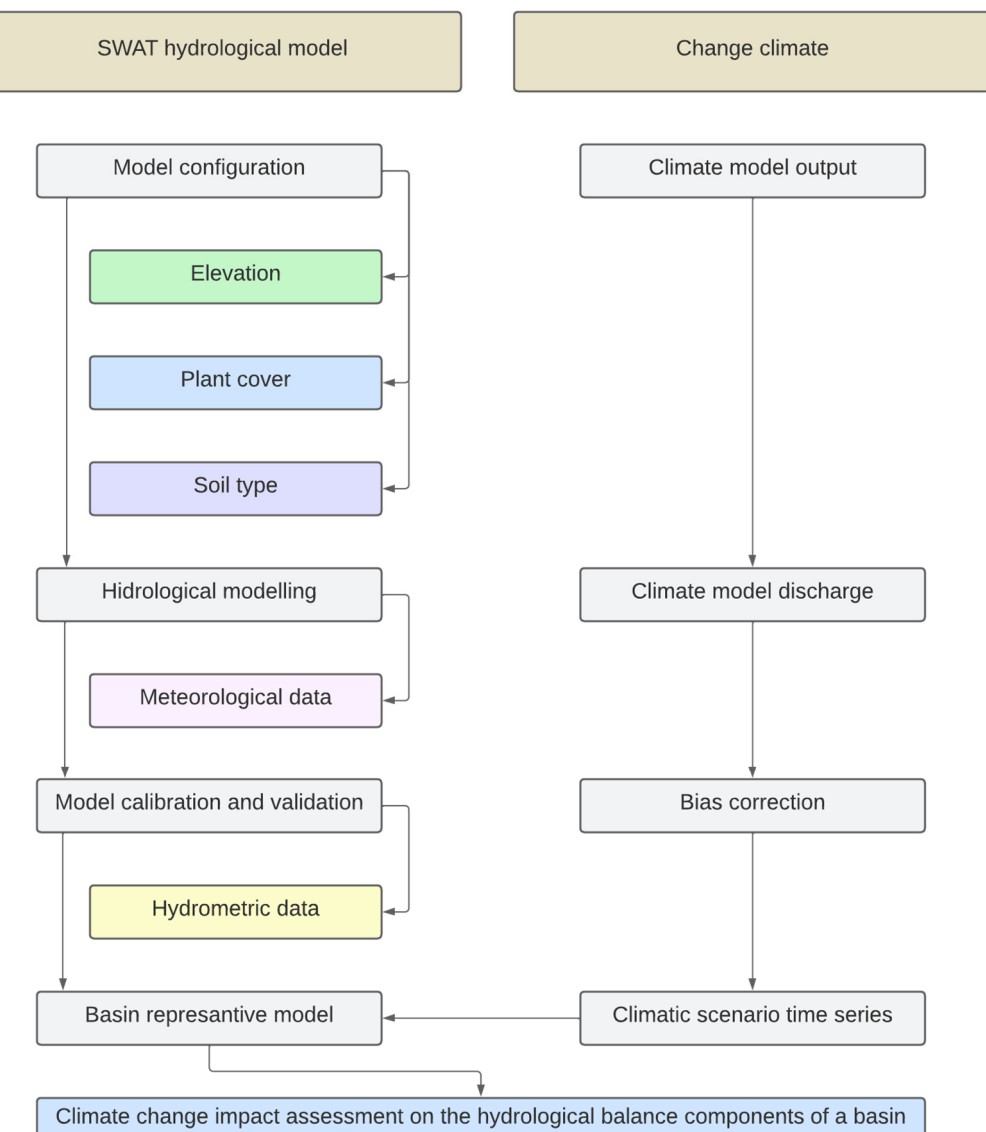

**Figure 3.** Methodology to assess the impact of climate change on hydrological balance.

**Table 1.** Input data for the SWAT hydrological model.

| Type of Data | Description/Scale | Source |
|---|---|---|
| Digital Elevation Model | DEM Resolution: 15 m | National Institute of Statistics and Geography [42] |
| Plant cover | Series IV Land Use and Vegetation Chart: 1:250,000 scale | National Institute of Statistics and Geography [43] |
| Soil type | Digital soil classification: 1:60,000 scale | Digital Soil Mapping Techniques [44] |
| Precipitation and temperature | 1970–1980 (daily) | Servicio Meteorológico Nacional [45] |
| Stream flow | 1970–1980 (monthly) | Comisión Nacional del Agua [46] |

### 2.3. Configuration of Model

The SWAT model was implemented using the QSWAT3 plugin within the graphical interface of the open-source program QGIS version 3.16.8 [47]. The delineation of the watershed was carried out in five stages [48]: (1) configuration of the 15 m resolution DEM; (2) definition of the stream network based on a flow accumulation area of 50 km$^2$, employing the eight-direction flow (D8) algorithm [49]; (3) identification of inputs and outputs of sub-watersheds from the intersections of watercourses; (4) location of the watershed's outlet; and (5) selection, definition, and calculation of the parameters of the sub-watersheds. The outlet points are generated automatically by the SWAT model for each sub-watershed. Furthermore, it allows for the manual addition of user-defined points corresponding to stream flow measurement sites in the water courses. Therefore, according to the delineation, the MRB covers an area of 6559.20 km$^2$ and was subdivided into 79 sub-watersheds.

Subsequently, each sub-watershed was subdivided into homogeneous areas termed Hydrological Response Units (HRU) based on the overlay of thematic layers of vegetation cover, soil type, and terrain slope; the latter was classified into two intervals, less than 5% and greater than 5%. The vegetation cover was derived from the spatial distribution of the Land Use and Vegetation Map Series IV from INEGI [43]. The characterization of the main soil groups was obtained from field profile surveys, and from the spatial distribution resulting from Digital Soil Mapping (DSM) techniques [44]. Additionally, to enhance the efficiency of the SWAT model in estimating runoff production, it is recommended to discretize the HRUs based on a threshold value of 5%/5%/5% for each thematic layer [50]. Finally, the SWAT model identified a total of 1239 HRUs within the MRB.

Meteorological data provide the amount of energy and water that controls the water balance components and highlight their relative importance in the basin [48]. To do this, the SWAT model uses the daily climate variables of precipitation, maximum and minimum temperature, solar radiation, average wind speed, and average relative humidity. These variables could be entered or estimated from the WXGEN climate generator, which works based on the monthly statistics of each variable. The MRB has twelve meteorological stations that record precipitation and temperature daily [45]. For each meteorological station, the other variables were estimated monthly; wind speed and solar radiation were generated from the information available from the Climate Forecast Reanalysis System (CFSR) data [51]. The maximum rainfall intensity in 30 min was obtained using Chen's method [52,53], and the dew point temperature was based on the minimum temperature [54]. Therefore, due to the limited and scarce availability of relative humidity, wind speed, and solar radiation variables to calculate the ETP, the HS method was used, which has shown good performance in semi-arid areas [55].

### 2.4. Model Calibration and Validation

The first two years were established as warm up to adjust the initial moisture content of the soil [56]. Subsequently, the calibration (1972–1976) and validation (1977–1980) period were selected based on the available current streamflow information.

The calibration of the SWAT model consists of estimating values of a set of parameters that minimize the difference between the observed and simulated data [57]. SWATCUP (SWAT Calibration and Uncertainty Program) software was used, which provides a decision-making framework through a semi-automatic approach for sensitivity analysis, calibration, validation, and information certainty [58]. In this research, the Sequential Uncertainty Adjustment algorithm (SUFI-2) was used by selecting parameters associated with the processes of runoff production, groundwater flow, the definition of the HRU, and soil type [59], as shown in Table 2. The SWATCUP was executed by defining the objective function, the initial range, and the change method of each parameter. The latter could be determined by the methods of relative change (r\_), replace (v\_), and absolute (a\_). The term "r" is used for the relative adjustment of a parameter within a given range; the term "v" directly replaces the parameter value with an assigned value, and the term "a" adds a value to the parameter [58].

**Table 2.** SWAT model calibration parameters.

| Parameter | Description | Method |
|---|---|---|
| ALPHA_BF | Baseflow alpha factor (day) | Replace |
| ALPHA_BNK | Baseflow alpha factor for bank storage (day) | Replace |
| CH_K1 | Hydraulic conductivity in alluvium (mm h$^{-1}$) | Replace |
| CH_K2 | Effective hydraulic conductivity in the main channel (mm h$^{-1}$) | Replace |
| CN2 | SCS runoff curve number (dimensionless) | Add |
| DEP_IMP | Depth of impervious layer (mm) | Replace |
| EPCO | Plant uptake compensation factor (dimensionless) | Replace |
| ESCO | Soil evaporation compensation factor (dimensionless) | Replace |
| GW_DELAY | The time interval for recharge of the aquifer (day) | Replace |
| GW_REVAP | Groundwater revap coefficient (dimensionless) | Replace |
| GWQMN | The threshold depth of water in the shallow aquifer required for return flow to occur (mm) | Replace |
| HRU_SLP | Average slope steepness (m/m) | Relative |
| LAT_TTIME | Lateral flow travel time (day) | Replace |
| RCHRG_DP | Deep aquifer percolation function (dimensionless) | Replace |
| SLSOIL | Slope length for lateral subsurface flow (m) | Relative |
| SLSUBBSN | Average slope length (m) | Relative |
| SOL_AWC | Soil available water storage capacity (mm mm$^{-1}$) | Relative |
| SOL_K | Soil hydraulic conductivity (mm h$^{-1}$) | Relative |
| SOL_Z | Soil depth (mm) | Relative |
| SURLAG | Surface runoff lag coefficient (day) | Replace |

Finally, the validation of the model determined the precision of surface runoff production during the validation period without making any additional adjustments to parameters calibrated using SWATCUP.

The SWAT model's performance was evaluated using the evaluation metrics recommended by Moriasi et al. [60]: coefficient of determination ($R^2$), Nash–Sutcliffe efficiency (*NSE*), and percentage bias (*PBIAS*), according to Equations (4)–(6).

$$R^2 = \frac{\left[\sum_{i=1}^{n}\left(Q_{obs}(i) - \overline{Q_{obs}}\right)\left(Q_{sim}(i) - \overline{Q_{sim}}\right)\right]^2}{\sum_{i=1}^{n}\left(Q_{obs}(i) - \overline{Q_{obs}}\right)^2 \sum_{i=1}^{n}\left(Q_{sim}(i) - \overline{Q_{sim}}\right)^2} \tag{4}$$

$$NSE = 1 - \frac{\sum_{i=1}^{n}\left(Q_{obs}(i) - Q_{sim}(i)\right)^2}{\sum_{i=1}^{n}\left(Q_{obs}(i) - \overline{Q_{obs}}\right)^2} \tag{5}$$

$$PBIAS = \frac{\sum_{i=1}^{n}\left(Q_{obs}(i) - Q_{sim}(i)\right)}{\sum_{i=1}^{n}Q_{obs}(i)} \times 100 \tag{6}$$

where $Q_{obs}$ is the observed flow, $Q_{sim}$ is the simulated flow in a month $i$, $\overline{Q_{obs}}$ is the average of the observed flow, and $\overline{Q_{sim}}$ is the average of the simulated flow. $R^2$ varies from 0 to 1, where 0 indicates no correlation and 1 corresponds to perfect correlation and lower error variance. *NSE* can vary from $-\infty$ to 1, where values $\leq 0$ show that the model is not reliable and values closer to 1 indicate a perfect fit between the observed and simulated data. The best value of *PBIAS* is 0, the underestimation and overestimation of the model correspond to positive and negative values, respectively.

*2.5. Global Climate Models*

Global Climate Models (GCMs) are the tools most commonly used for assessing the potential impacts of climate change because they provide valuable information, although they generate systematic biases, mainly in rugged relief areas [61]. The Coupled Model Inter-Comparison Project (CMIP) is a global climate modeling framework that has allowed for a better understanding of the climate system and knowledge of different scenarios of greenhouse gas (GHG) concentrations that are mainly due to human activities; it is currently in phase six (CMIP6) [62,63]. However, despite the progress of GCMs on a global scale, they still exhibit a more significant bias in precipitation than in surface air temperature compared to climate models developed in previous phases [64].

The climate scenarios of the GCMs in the IPCC Sixth Assessment Report (AR6) contain four Shared Socioeconomic Pathways (SSP), representing different socioeconomic developments and atmospheric GHG concentration pathways. The first scenario, SSP126, corresponds to sustainable socioeconomic development by optimizing material and energy resources, with a radiative forcing of 2.6 W m$^{-2}$. The second scenario, SSP245, corresponds to moderate development, in which countries establish climate change mitigation and adaptation measures with a radiative forcing of 4.5 W m$^{-2}$. The third, SSP370, corresponds to moderate to high development, wherein countries stop implementing climate change mitigation and adaptation measures because of their need to increase energy and food security, with a radiative forcing of 7.0 W m$^{-2}$. Finally, the fourth scenario, SSP585, corresponds to socioeconomic development driven primarily by fossil fuels, leaving aside the application of any adaptation and mitigation measures to climate change, with a radiative forcing of 8.5 W m$^{-2}$. All of the above scenarios are for the year 2100 [9,65]. Scenarios SSP245 and SSP585 can cover a wide range of possibilities, from low- to medium- and high-emission scenarios. Therefore, they were selected to assess the impact of climate change on the hydrological balance of the MRB.

Table 3 shows the GCMs used in this study for the historical (1970–1980) and future (2020–2099) scenarios of precipitation, maximum temperature, and minimum temperature, with data obtained through the https://esgf-node.llnl.gov/projects/cmip6/ (accessed on 15 July 2022) data platform of the Earth System Grid Federation (ESGF).

**Table 3.** Information about the Global Climate Models used.

| Number | Model Name | Country | Institution | Resolution (km) |
|--------|------------|---------|-------------|-----------------|
| 1 | CNRM-CM6-1 | France | CNRM-CERFACS | 250 |
| 2 | MRI-ESM2-0 | Japan | MRI | 100 |
| 3 | ACCESS-ESM1-5 | Australia | CSIRO | 250 |
| 4 | MIROC6 | Japan | MIROC | 250 |
| 5 | MPI-ESM1-2-LR | Germany | MPI-M | 250 |
| 6 | HadGEM3-GC31-LL | United Kingdom | MOHC | 250 |
| 7 | BCC-CSM2-MR | China | BCC | 100 |
| 8 | CanESM5 | Canada | CCCma | 500 |
| 9 | GFDL-CM4 | United States | NOAA-GFDL | 100 |
| 10 | GFDL-ESM4 | United States | NOAA-GFDL | 100 |

Due to the abundance of GCMs in CMIP6, it is challenging to include all of them in climate change research. Therefore, for practical purposes, a climate model or a small ensemble is selected, which can represent the past and present climate of the area of interest and provide an adequate prediction of the future [66]. Furthermore, it is essential to note that its purpose is to generate results to cover a wide range of uncertainty in climate change

scenarios and provide the most significant possible information, which can be crucial for evaluation in the context of climate change and decision-making.

### 2.6. Bias Correction and Climate Model Selection

Frequently, GCMs involve a considerable bias in precipitation and temperature, mainly due to the influence of relief [67,68]. Therefore, bias correction is necessary to minimize the discrepancy between the time series of climate model analysis variables and observed data [69], in order to model the components of hydrological balance properly.

The bias correction was carried out with the CMhyd (Climate Model data for hydrological modeling) tool, which has several correction methods that allow for obtaining representative simulated climate data, according to the location and historical record of the precipitation variables and temperature, for integration into the hydrological modeling of the basin [70]. Bias correction methods are typically designed to modify the climate model's mean, distribution, and variance [71]. Table 4 shows the bias correction methods used in the present study through the CMhyd tool.

**Table 4.** Bias correction methods used for precipitation and temperature.

| Method | Description |
| --- | --- |
| Linear Scaling (LS) | Correction of monthly values using a multiplicative (for precipitation) or an additive (for temperature) factor based on differences between simulated and observed data |
| Distribution Mapping (DM) | Correction method is undertaken by shifting the gamma distribution (for precipitation) or Gaussian distribution (for temperature) using a transfer function |

Given the variability between the values derived from the different GCMs available from the CMIP6, it is necessary to compare them to select a climate model that adequately explains the behavior of each variable in the time series relative to the reference values.

The equations of the evaluation metrics of the different GCMs used in this study are shown below: correlation coefficient ($r$), Root Mean Square Error ($RMSE$), and standard deviation ($SD$).

$$r = \frac{\sum_{i=1}^{n}(x_i - x_m)(y_i - y_m)}{\sqrt{\sum_{i=1}^{n}(y_i - y_m)^2}\sqrt{\sum_{i=1}^{n}(x_i - x_m)^2}} \tag{7}$$

$$RMSE = \sqrt{\frac{\sum_{i=1}^{n}(y_i - x_i)}{n}} \tag{8}$$

$$SD = \sqrt{\frac{1}{n}\sum_{i=1}^{n}(y_i - y_m)^2} \tag{9}$$

where $x$ is the average of the observed precipitation and temperature of the month $i$, $x_m$ is the average of the precipitation and temperature observed during the period of analysis, $y$ is the average of the corrected precipitation and temperature of the month $i$, $y_m$ is the average of corrected precipitation and temperature during the period of analysis, and $n$ is the number of months with data for precipitation and temperature.

Additionally, the Taylor diagram was also used to evaluate the capability of the different climate models, since it allows for visualizing the correlation coefficient ($r$), Root Mean Square Error ($RMSE$), and standard deviation ($SD$) [72]. Therefore, it can be used to measure the precision and consistency of each variable [73].

### 3. Results and Discussion

### 3.1. Model Calibration and Validation

The MRB is located in the upper part of the Rio Balsas Hydrological Region (BRHR) and was delimited from the Mariscala Hydrometric Station (MHS). Currently, the BRM presents a high level of alteration in the discharge of surface runoff, caused mainly by the

change in land use from temperate and tropical forests to agricultural, livestock and urban use; furthermore, this basin belongs to the high areas of the BRHR, which together produce at least 25% of the drinking water consumed in the metropolitan area of Mexico City [74]. The MHS was neglected and presents a discontinuous record of stream flow information, having the most complete record for the period from 1970 to 1980 and ceasing to operate in 1995. For this reason, the period from 1970 to 1980 was used to develop this research.

The first two years were the warm-up to adjust the initial soil content. Table 5 shows the sensitivity analysis results and the set of parameters adjusted during calibration obtained using the SWATCUP software's SUFI-2 algorithm.

**Table 5.** Calibration of the parameters that influence the production of surface runoff in the MBR.

| Ranking | Parameter | Default Range | | Calibrated Range | | Fitted Value | Type |
| --- | --- | --- | --- | --- | --- | --- | --- |
| | | Lower Limit | Upper Limit | Lower Limit | Upper Limit | | |
| 1 | LAT_TTIME | 0.00 | 180.00 | 0.00 | 21.11 | 7.72 | hru |
| 2 | HRU_SLP | −0.20 | 0.20 | −0.23 | −0.15 | −0.19 | hru |
| 3 | SOL_K | −0.80 | 0.80 | −0.48 | −0.36 | −0.42 | sol |
| 4 | RCHRG_DP | 0.00 | 1.00 | 0.03 | 0.09 | 0.06 | gw |
| 5 | CN2 | −15.00 | 15.00 | −9.51 | −6.90 | −8.21 | mgt |
| 6 | SOL_Z | −0.35 | 0.35 | 0.25 | 0.39 | 0.32 | sol |
| 7 | SLSOIL | −0.10 | 0.10 | −0.07 | −0.02 | −0.10 | hru |
| 8 | CH_K2 | 5.00 | 130.00 | 9.92 | 27.38 | 18.65 | rte |
| 9 | GW_REVAP | 0.02 | 0.20 | 0.08 | 0.13 | 0.11 | gw |
| 10 | ESCO | 0.00 | 1.00 | 0.23 | 0.34 | 0.28 | hru |
| 11 | ALPHA_BNK | 0.00 | 1.00 | 0.50 | 0.62 | 0.56 | rte |
| 12 | SLSUBBSN | −0.30 | 0.30 | 0.06 | 0.10 | 0.09 | hru |
| 13 | SURLAG | 0.00 | 10.00 | 6.75 | 8.50 | 7.62 | bsn |
| 14 | GW_DELAY | 0.00 | 500.00 | 236.74 | 331.27 | 284.00 | gw |
| 15 | EPCO | 0.00 | 1.00 | 0.44 | 0.57 | 0.50 | hru |
| 16 | GWQMN | 0.00 | 5000.00 | 3031.08 | 3614.05 | 3322.56 | gw |
| 17 | CH_K1 | 0.00 | 300.00 | 3.68 | 79.88 | 41.78 | sub |
| 18 | ALPHA_BF | 0.00 | 1.00 | 0.57 | 0.96 | 0.76 | gw |
| 19 | SOL_AWC | −0.35 | 0.35 | 0.04 | 0.08 | 0.06 | sol |
| 20 | DEP_IMP | 0.00 | 6000.00 | 5033.51 | 5654.88 | 5344.20 | hru |

During the execution of the SWATCUP, three iterations were carried out with 1500 simulations each, with the NSE objective function. Therefore, according to the sensitivity analysis results and the attributes of the MRB, the parameters LAT_TTIME, HRU_SLP, SOL_K, RCHRG_DP, and CN2 significantly influence surface runoff production.

The average surface runoff observed at the MRB stream flow measurement hydrometric station was 24.38 m$^3$ s$^{-1}$, while hydrological modeling gave a value of 23.76 m$^3$ s$^{-1}$. The performance of the SWAT model showed good results in simulating surface runoff production during the calibration stage ($R^2$ = 0.86; *NSE* = 0.83; *PBIAS* = 13.93%) and validation stage ($R^2$ = 0.78; *NSE* = 0.76; *PBIAS* = −17.03%) as shown in Figure 4. Therefore, it meets the evaluation criteria [60]. In general, the model simulation result slightly underestimates the high flows that occur during the rainy season.

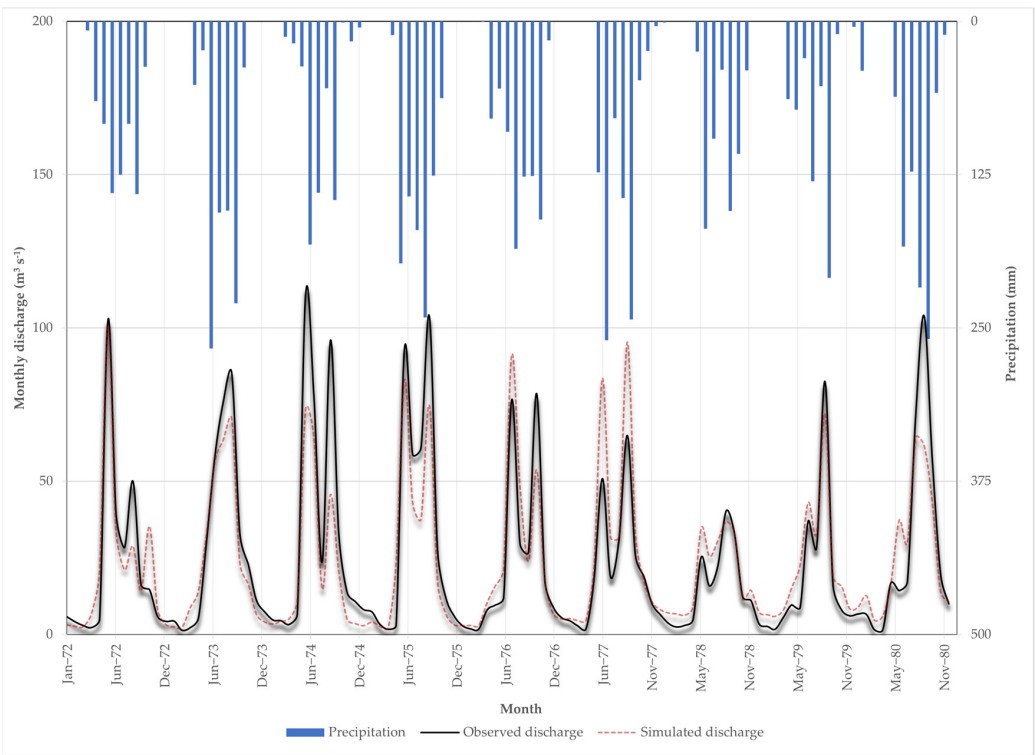

**Figure 4.** Hydrograph of the MRB in monthly steps.

Based on the hydrological modeling of the MRB, the baseline corresponded to the values of the hydrological balance components from 1972 to 1980. Table 6 shows the monthly precipitation values (PCP), surface runoff (SR), lateral flow (FL), evapotranspiration (ET), water yield (WY), and potential evapotranspiration (PET). Generally, the annual and monthly averages of a basin's different hydrological balance components show a direct relationship between precipitation and its other components. According to Desai et al. [75], the largest volume is released to the atmosphere through evapotranspiration, mainly due to the association between the spatial distribution of plant cover and temperature; in the particular case of the MRB, this accounts for 65.68% (481.80 mm) of precipitation. WY refers to the annual amount of water that leaves the basin, with a value of 13.63% (99.97 mm). Finally, the value of SR is only positive during the rainy season because the MRB is characterized by hydrological droughts during the third part of the year, with which flow production decreases considerably [75].

**Table 6.** Components of the monthly hydrological balance of the MRB.

| Month | PCP (mm) | SR (mm) | FL (mm) | ET (mm) | WY (mm) | PET (mm) |
|---|---|---|---|---|---|---|
| January | 5.88 | 0.02 | 0.37 | 5.92 | 1.32 | 121.2 |
| February | 2.34 | 0.00 | 0.34 | 4.70 | 1.19 | 132.02 |
| March | 4.79 | 0.00 | 0.32 | 5.01 | 1.16 | 175.57 |
| April | 33.71 | 0.14 | 2.13 | 24.94 | 2.99 | 184.88 |
| May | 80.21 | 0.94 | 6.18 | 51.48 | 7.52 | 187.79 |
| June | 155.82 | 4.04 | 16.82 | 84.23 | 20.09 | 154.89 |
| July | 127.71 | 2.60 | 14.95 | 82.59 | 17.15 | 156.14 |
| August | 106.41 | 2.18 | 11.42 | 74.09 | 13.71 | 154.51 |
| September | 149.91 | 5.25 | 16.55 | 78.75 | 20.62 | 137.94 |
| October | 43.24 | 0.64 | 7.12 | 40.76 | 8.42 | 133.52 |
| November | 19.49 | 0.46 | 2.71 | 21.36 | 4.05 | 117.93 |
| December | 4.03 | 0.00 | 0.74 | 7.97 | 1.75 | 112.92 |

### 3.2. Assessment and Selection of Climate Models

The selection of the GCM consisted of the analysis of the precipitation, maximum temperature, and minimum temperature monthly time series for the historical period 1970–1980. This was based on the results of the evaluation metrics, correlation coefficient (*r*), Root Mean Square Error (*RMSE*), and standard deviation (*SD*), as well as the visualization of the Taylor diagram prepared in the R−Studio software version 1.4.1106 with the package plotrix.

Table 7 shows the evaluation metrics for maximum temperature (Tmax), minimum temperature (Tmin), and precipitation (Prec) metrics for the ten GCMs used in this study. In general, the maximum and minimum temperature showed a better fit compared to precipitation, and the Linear Scaling method (LS) obtained better results compared to the Distribution Mapping method (DM).

The minimum temperature (Tmax) and maximum temperature (Tmin) have a value of r greater than 0.85, except in the case of the GCM GFDL-CM4: this value is less than or equal to 0.80 in precipitation. The value of the RMSE varies for the maximum temperature from 1.74 to 3.30 °C month$^{-1}$, the minimum temperature from 1.42 to 2.98 °C month$^{-1}$, and the precipitation from 45.80 to 77.71 mm month$^{-1}$.

**Table 7.** Evaluation metrics of GCMs with Linear Scaling method.

| GCM | Method of Bias Correction | r | | | RMSE | | | SD | | |
|---|---|---|---|---|---|---|---|---|---|---|
| | | Tmax | Tmin | Prec | Tmax | Tmin | Prec | Tmax | Tmin | Prec |
| CNRM-CM6-1 | LS | 0.90 | 0.91 | 0.74 | 1.75 | 1.61 | 52.04 | 3.80 | 3.75 | 66.53 |
| | DM | 0.89 | 0.91 | 0.63 | 1.82 | 1.60 | 66.67 | 3.83 | 3.75 | 79.12 |
| MRI-ESM2-0 | LS | 0.90 | 0.92 | 0.73 | 1.74 | 1.51 | 53.73 | 3.82 | 3.73 | 68.88 |
| | DM | 0.90 | 0.91 | 0.61 | 1.81 | 1.59 | 68.60 | 3.85 | 3.76 | 80.43 |
| ACCESS-ESM1-5 | LS | 0.86 | 0.93 | 0.70 | 2.13 | 1.47 | 58.69 | 3.97 | 3.72 | 75.14 |
| | DM | 0.86 | 0.91 | 0.60 | 2.15 | 1.58 | 77.71 | 3.98 | 3.77 | 93.62 |
| MIROC6 | LS | 0.86 | 0.93 | 0.70 | 2.08 | 1.42 | 56.90 | 3.92 | 3.70 | 70.99 |
| | DM | 0.88 | 0.91 | 0.60 | 1.92 | 1.60 | 71.49 | 3.84 | 3.77 | 83.83 |
| MPI-ESM1-2-LR | LS | 0.88 | 0.92 | 0.80 | 1.91 | 1.56 | 45.80 | 3.90 | 3.76 | 65.33 |
| | DM | 0.89 | 0.91 | 0.71 | 1.88 | 1.61 | 57.20 | 3.89 | 3.78 | 74.21 |
| HadGEM3-GC31-LL | LS | 0.88 | 0.89 | 0.70 | 1.89 | 1.77 | 57.64 | 3.87 | 3.82 | 73.58 |
| | DM | 0.87 | 0.90 | 0.61 | 2.03 | 1.75 | 69.78 | 3.93 | 3.81 | 82.01 |
| BCC-CSM2-MR | LS | 0.87 | 0.92 | 0.65 | 2.03 | 1.53 | 65.72 | 3.92 | 3.73 | 80.63 |
| | DM | 0.88 | 0.91 | 0.62 | 1.90 | 1.59 | 70.28 | 3.86 | 3.75 | 84.56 |
| CanESM5 | LS | 0.89 | 0.93 | 0.76 | 1.87 | 1.44 | 50.53 | 3.87 | 3.72 | 69.11 |
| | DM | 0.89 | 0.93 | 0.71 | 1.87 | 1.48 | 57.10 | 3.87 | 3.74 | 74.04 |
| GFDL-CM4 | LS | 0.61 | 0.66 | 0.72 | 3.24 | 2.98 | 54.66 | 2.93 | 3.23 | 70.31 |
| | DM | 0.59 | 0.67 | 0.60 | 3.30 | 2.95 | 74.34 | 3.01 | 3.20 | 88.88 |
| GFDL-ESM4 | LS | 0.88 | 0.92 | 0.72 | 1.94 | 1.51 | 54.37 | 3.89 | 3.74 | 68.71 |
| | DM | 0.87 | 0.92 | 0.59 | 2.02 | 1.52 | 74.07 | 3.92 | 3.75 | 86.62 |

The Taylor diagram offers a concise statistical summary of the evaluation metrics (*r*, *RMSE*, and *SD*) between the observed and corrected climate variables (precipitation and temperature) of the different GCMs. Figure 5 shows the Taylor diagram, where both the *x*-axis and the *y*-axis indicate the *SD*; the black dashed lines represent the *r* between the observed and corrected variable; the *RMSE* of the corrected variable is proportional to the distance from the *x*-axis identified as "observation" (cyan contours) and the *SD* is proportional to the radial distance from the point of origin (blue contours).

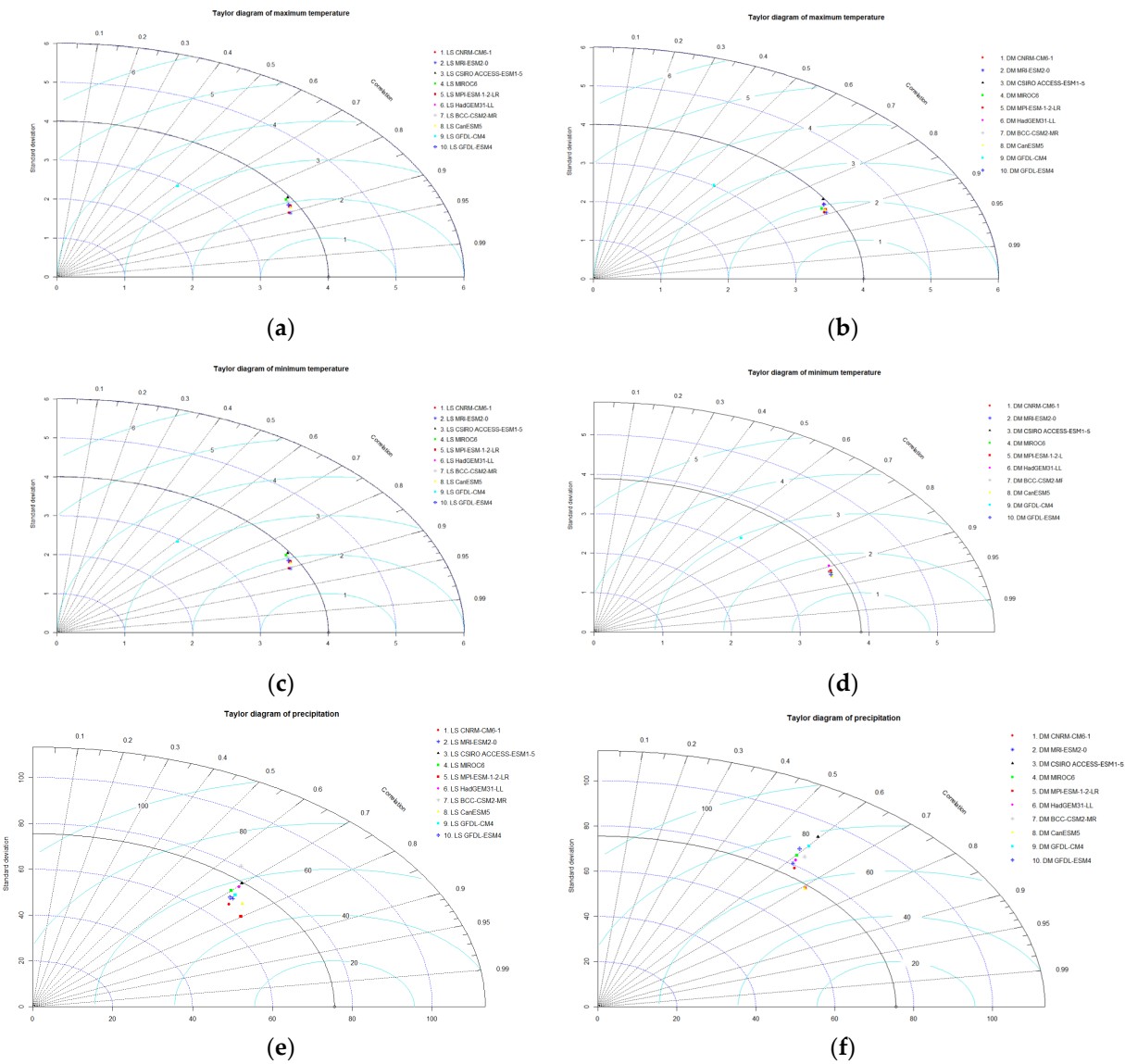

**Figure 5.** Taylor diagrams: (**a**) maximum temperature with the LS method; (**b**) maximum temperature with the DM method; (**c**) minimum temperature with the LS method; (**d**) minimum temperature with the DM method; (**e**) precipitation with the LS method; (**f**) precipitation with the DM method.

With the LS method, two climate models were highly accurate for the maximum temperature, with *r* values greater than 0.90 and an *RMSE* lower than 1.75 °C month$^{-1}$; the most accurate GCM was MRI-ESM2-0. Seven climate models were also highly precise for the minimum temperature, with *r* values above 0.93 and an *RMSE* lower than 1.50 °C month$^{-1}$; the most accurate GCM was MIROC6. However, in the precipitation variable, the GCMs yielded r values between 0.7 and 0.8, with the most accurate GCM being MPI-ESM-1-2-LR.

Likewise, with the DM method, two climate models were highly accurate for the maximum temperature, with r values greater than 0.89 and an RMSE lower than 1.85 °C month$^{-1}$; similarly, the most accurate GCM was MRI-ESM2-0. In addition, seven climate models were highly precise for the minimum temperature, with *r* values varying from 0.86 to 0.90 and an RMSE lower than 1.60 °C month$^{-1}$; the most accurate GCM was MIROC6. In the same way, the precipitation variable of the GCMs yielded *r* values between 0.6 and 0.7, with the most accurate GCM being CanESM5.

In general, the GCM MPI-ESM-1-2-LR achieved the best performance according to the evaluation metrics between the time series of the variables analyzed, compared to the values observed in the historical period through the LS method of bias correction. Therefore, we used the information from this GMC for the hydrological modeling of the MRB. Likewise, it was used as the basis for deriving the climate scenarios SSP245 and SSP585, which correspond to radiative forcing levels of 4.5 W m$^{-2}$ and 8.5 W m$^{-2}$, respectively, for the year 2100.

### 3.3. Impact on Precipitation and Temperature

In the MRB, the baseline mean annual precipitation is 733.55 mm, and the mean annual temperature is 19.43 °C. However, climate change scenarios foresee changes in both variables. The SSP245 scenario of GCM MPI-ESM-1-2-LR predicts a decrease in precipitation and a rise in the mean annual and monthly temperature. In the case of precipitation, a reduction of 44.44 mm is projected in the short term (2025–2049), 76.42 mm in the medium term (2050–2074), and 83.71 mm in the long term (2074–2099), while temperature will increase by 1.60 °C in the short term (2025–2049), 2.31 °C in the medium term (2050–2074), and 2.57 °C in the long term (2075–2099). Figure 6 illustrates the monthly behavior of both variables for the SSP245 scenario, showing a decrease in precipitation mainly during the rainy period and a rise in monthly mean temperature from November to April.

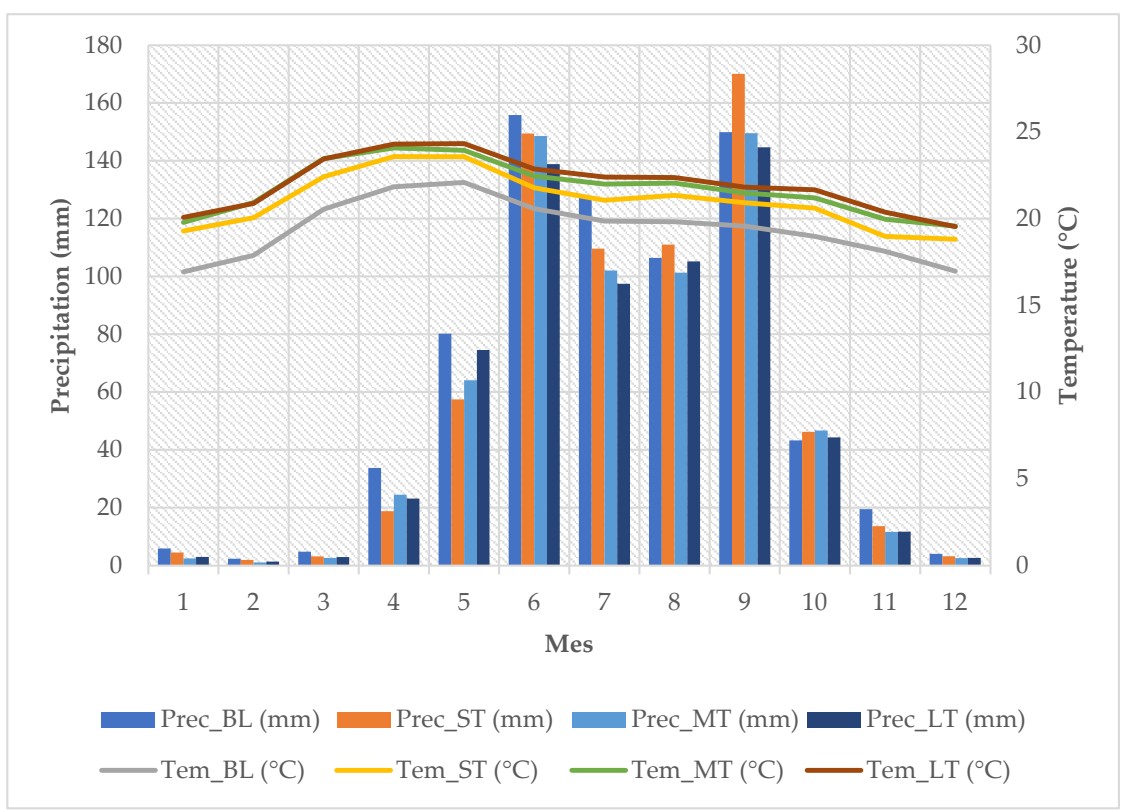

**Figure 6.** Behavior of precipitation and temperature under the SSP245 climate scenario of GCM MPI-ESM-1-2-LR.

The SSP585 scenario of GCM MPI-ESM-1-2-LR predicts a decrease in precipitation and an even more impactful increase in mean annual temperature. Regarding precipitation, a reduction of 54.29 mm is predicted in the short term (2025–2049), 115.27 mm in the medium term (2050–2074), and 225.83 mm in the long term (2074–2099), whereas temperature will increase by 1.79 °C in the short term (2025–2049), 3.00 °C in the medium term (2050–2074), and 4.77 °C in the long term (2075–2099). Furthermore, Figure 7 shows the monthly behavior of both variables for the SSP585 scenario.

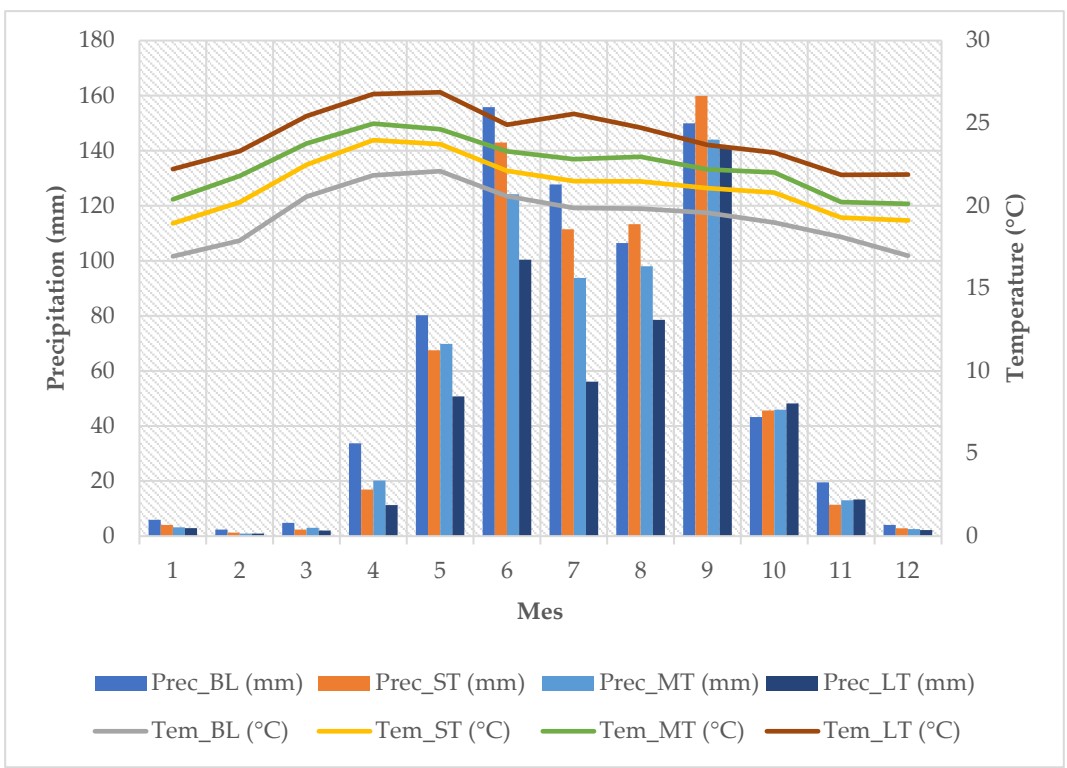

**Figure 7.** Behavior of precipitation and temperature under the SSP585 climate scenario of GCM MPI-ESM-1-2-LR.

Figure 8 shows the spatial distribution of changes in mean annual precipitation. The SSP245 climate scenario clearly shows a decrease in precipitation from 5% to 10% in the short term, and from 10% to 15% in the medium and long term. In contrast, in the SSP585 climate scenario, the percentage of change goes from 5% to 10% in the short term to 15% to 20% in the medium term, and, finally, from 30% to 35% in the long term.

Similarly, the spatial distribution of the mean annual temperature is shown in Figure 9. For the SSP245 climate scenario, the increase in temperature changes from 5% to 10% in the short term to 10% to 15% in the medium and long term although, in the latter, the sub-basins show increases of 15% to 20%. Under the SSP585 climate scenario, the percentage of change goes from 5% to 10% in the short term, from 15% to 20% in the medium term, and, finally, from 20% to 25% in the long term, with some sub-basins showing rises above 30%.

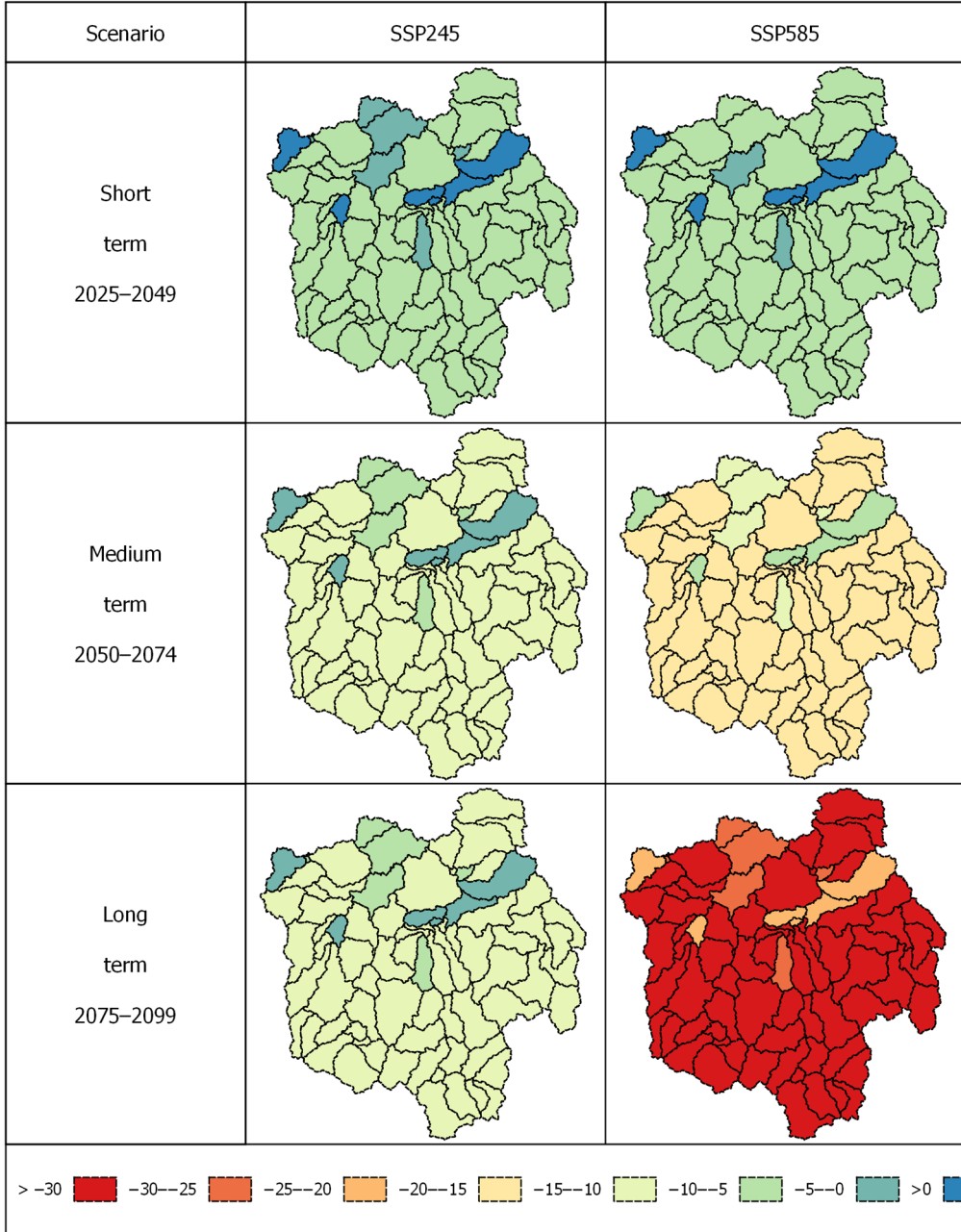

**Figure 8.** Percentage of change in precipitation in the MRB with GCM MPI-ESM-1-2-LR.

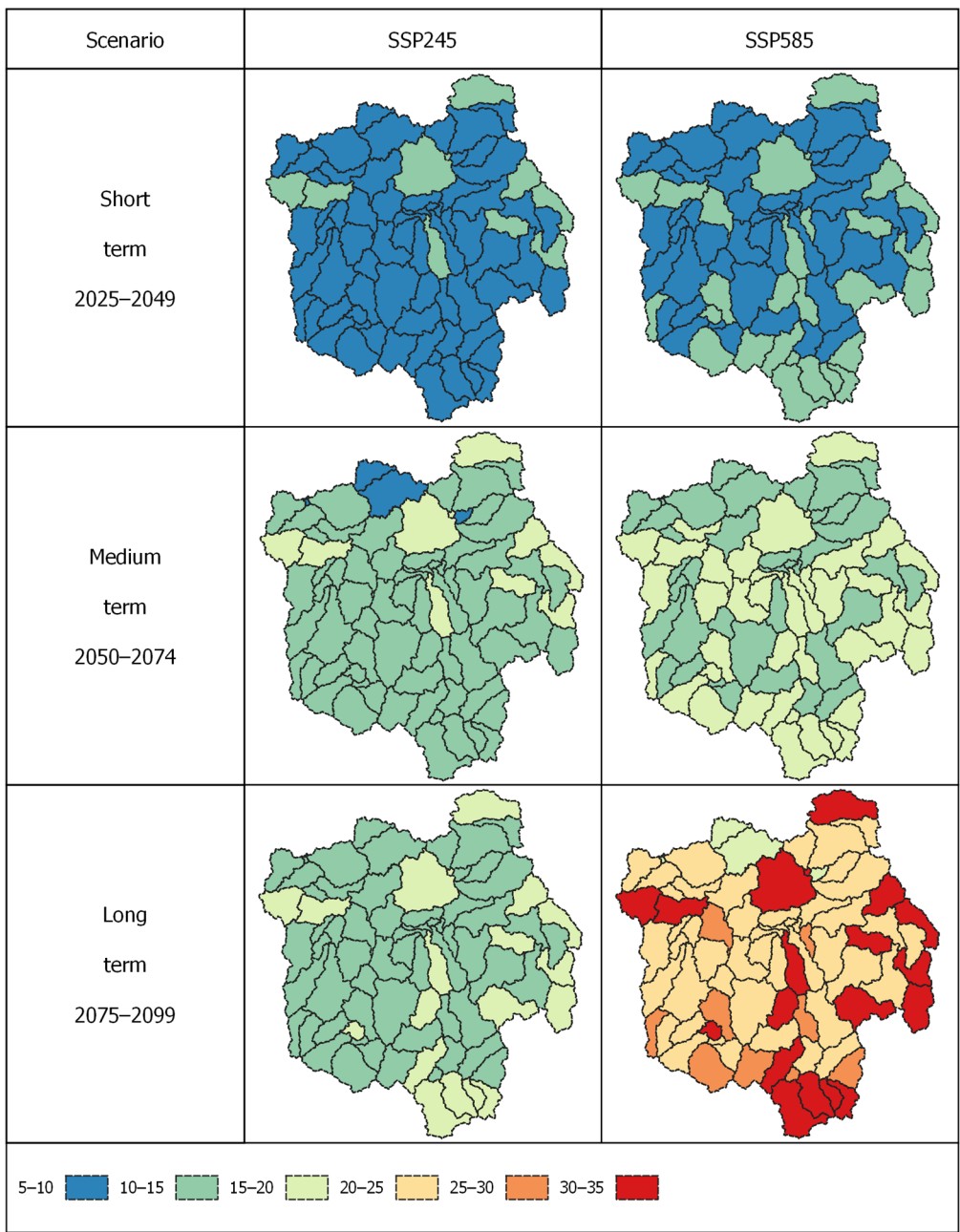

**Figure 9.** Percentage of change in temperature in the MRB with GCM MPI-ESM-1-2-LR.

*3.4. Impact on Hydrological Balance Components*

The impact of climate change was evaluated from the results of the hydrological modeling of the MRB (baseline), and the corrected projection of scenarios SSP245 and SSP585 of the GCM MPI-ESM-1-2-LR predicts that the spatial distribution of plant cover and soil, and their properties, remain constant. Therefore, the variation in the hydrological balance components corresponds only to the short-, medium-, and long-term effects of the change in precipitation and temperature.

The hydrological response of the MRB under the SSP245 and SSP585 climate scenarios under different time horizons and at the annual level of the hydrological balance components precipitation (PCP), potential evapotranspiration (PET), evapotranspiration (ET), and water yield (WY) are shown in Table 8.

**Table 8.** Changes in the components of hydrological balance in the MRB with GCM MPI-ESM1-2-LR.

| Month | Variable (mm) | Baseline (mm) | Short Term (2025–2049) | Medium Term (2050–2074) | Long Term (2075–2099) |
|---|---|---|---|---|---|
| | | | Value [Difference (%)] | Value [Difference (%)] | Value [Difference (%)] |
| SSP245 | PCP | 733.55 | 689.12 (−6.06) | 657.14 (−10.42) | 649.85 (−11.41) |
| | PET | 1769.31 | 1866.74 (5.51) | 1919.17 (8.47) | 1936.60 (9.45) |
| | ET | 481.80 | 586.58 (21.75) | 563.12 (16.88) | 560.69 (16.37) |
| | WY | 99.97 | 56.71 (−43.27) | 54.50 (−45.48) | 52.58 (−47.40) |
| SSP585 | PCP | 733.55 | 679.27 (−7.40) | 618.29 (−15.71) | 507.73 (−30.79) |
| | PET | 1769.31 | 1893.17 (7.00) | 1961.43 (10.86) | 2059.84 (16.42) |
| | ET | 481.80 | 579.94 (20.37) | 535.67 (11.18) | 441.79 (−8.30) |
| | WY | 99.97 | 55.58 (−44.40) | 48.78 (−51.21) | 38.98 (−61.01) |

The temperature rise was directly reflected in PET; this component increased significantly versus the baseline in both climate scenarios. Under the SSP245 climate scenario, PET would increase to 1866.74 mm year$^{-1}$ in the short term, 1919.17 mm year$^{-1}$ in the medium term, and 1936.59 mm year$^{-1}$ in the long term. Similarly, under the SSP585 climate scenario, PET would increase to 1893.17 mm year$^{-1}$ in the short term, to 1961.43 mm year$^{-1}$ in the medium term, and 2059.84 mm year$^{-1}$ in the long term.

The hydrological balance of the MRB indicates that a higher percentage of PCP will be consumed by ET instead of contributing to surface runoff, since the baseline ET was 481.80 mm year$^{-1}$. Still, it would increase by up to 20.36% under the climate change scenarios analyzed. Under the SSP245 climate scenario, PET would rise to 586.58 mm year$^{-1}$ in the short term, 563.13 mm year$^{-1}$ in the medium term, and 560.70 mm year$^{-1}$ in the long term. Similarly, under the SSP585 climate scenario, PET would increase to 1893.16 mm year$^{-1}$ in the short term, 535.68 mm year$^{-1}$ in the medium term, and 441.80 mm year$^{-1}$ in the long term. A decrease is observed along different time horizons, which is consistent with the decrease in precipitation, reflecting the exchange of energy in the soil–water–atmosphere processes [76].

Additionally, the future hydrological balance of the MRB points to a general decrease in WY, which will decrease by up to 47.40% and 61.01% under the climate scenarios SSP245 and SSP585, respectively. This is consistent with the figures reported in various studies [77–79]. Furthermore, the monthly decrease in WY coincides with the decrease in precipitation from May to October, as shown in Figures 10 and 11.

The decrease in precipitation and the increase in atmospheric evaporative demand due to the increasing warming of the planet toward the end of the 21st century will lead to a decrease in WY in the MRB and, consequently, lower availability of water resources in the basin [38,77]. This should be considered by resource managers in the development of management plans [79]. Furthermore, the increased frequency of droughts will impact the growth and development of the main crops, affecting their quality and yield [80]. This has a more significant effect in areas with semi-arid to arid climates, that is, in areas with less availability of water resources [71,77].

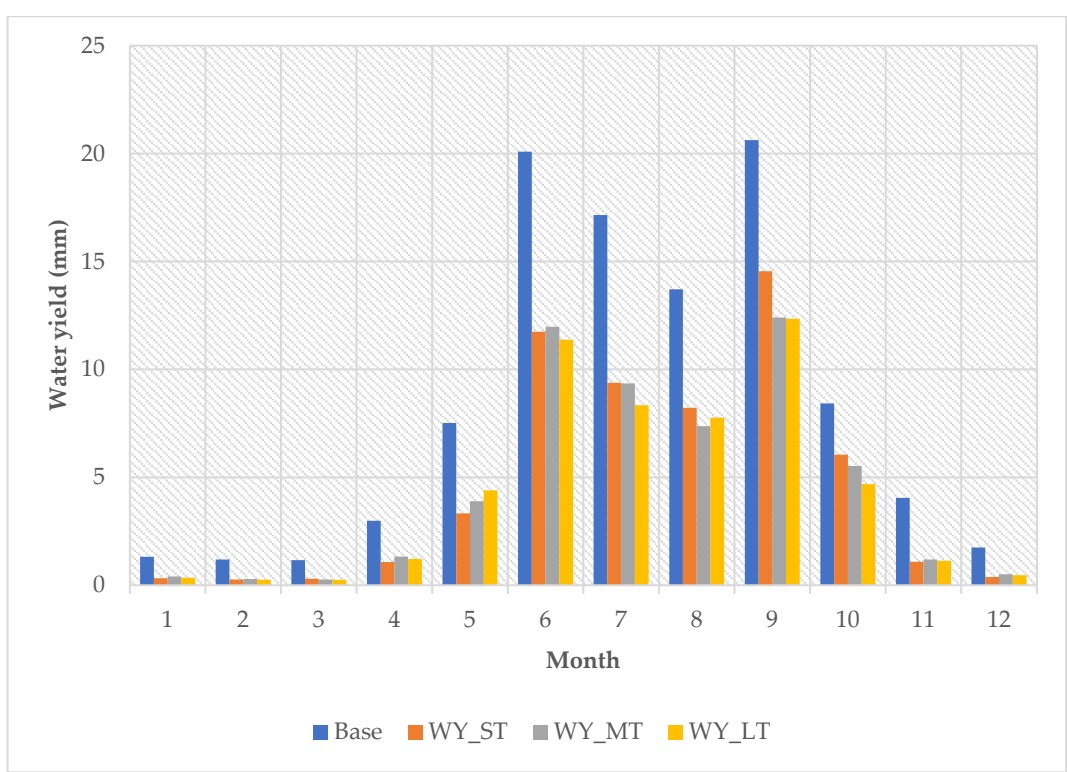

**Figure 10.** Behavior of water yield under the SSP245 climate scenario of GCM MPI-ESM-1-2-LR.

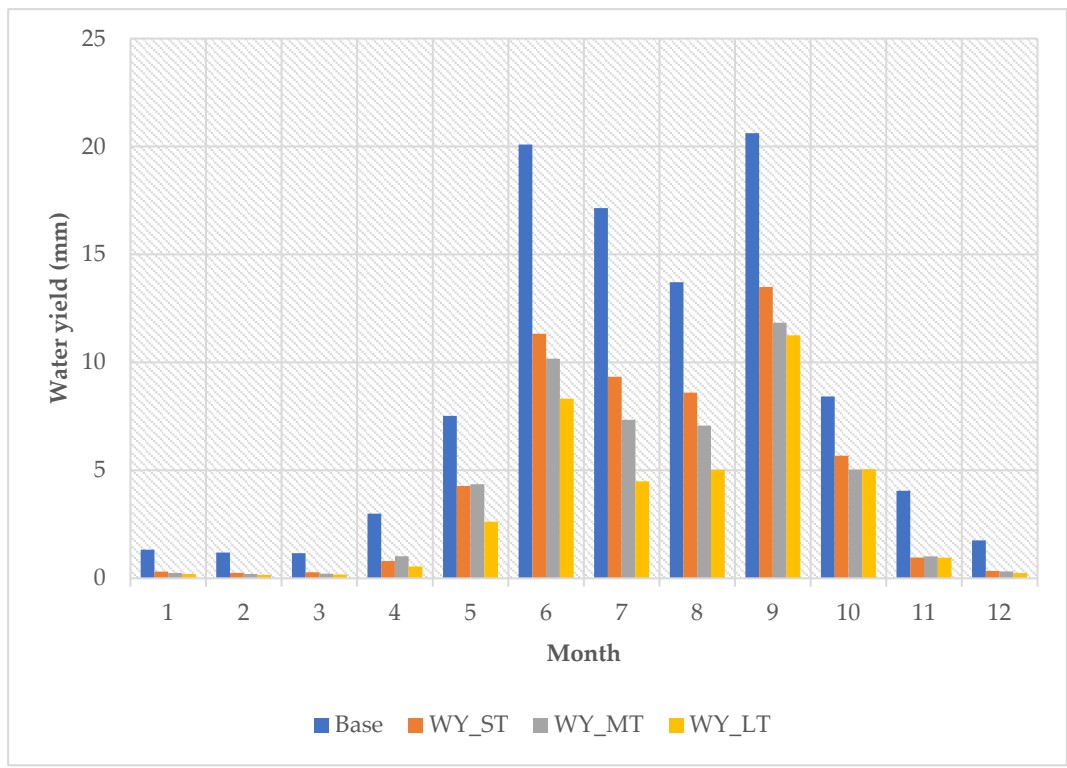

**Figure 11.** Behavior of water yield under the SSP585 climate scenario of GCM MPI-ESM-1-2-LR.

## 4. Conclusions

The SWAT hydrological model was calibrated and validated for the 1972–1980 period in the River Mixteco Basin (MRB). Based on evaluation metrics such as $R^2$, *NSE*, and *PBIAS*,

the SWAT model's performance is satisfactory for the given period of flow information availability. As a result, this model is recommended for predicting the potential impact of climate change on a basin scale.

The MPI-ESM1-2-LR climate model corrected by the linear scaling method adequately represented the behavior of precipitation and maximum and minimum temperatures. Therefore, it was used as the basis to analyze future scenarios of the impact of climate change in the MRB. This model projects a decrease in precipitation of 83.71 mm year$^{-1}$ to 225.83 mm year$^{-1}$ and a temperature rise between 2.57 °C and 4.77 °C under different time horizons for the climate scenarios SSP245 and SSP585, respectively.

In general, hydrological modeling predicts significant short-, medium-, and long-term changes in the components of the hydrological balance of the MRB under the climate scenarios SSP245 and SSP585, as they project a significant increase in the water demand of the plant cover, which is above the baseline, thereby affecting all components of the hydrological balance.

The MRB's water yield (WY) will decrease significantly under the climate scenarios SSP245 and SSP585. In the medium development scenario, with the establishment of mitigation and adaptation measures for climate change, a WY reduction of approximately 47.40% is expected. In contrast, in the most adverse scenario, i.e., socioeconomic development driven mainly by fossil fuels, WY is projected to experience a reduction of 61.01%.

Both scenarios evaluated project a marked reduction of the water available in the MRB. Therefore, designing and implementing short-, medium-, and long-term adaptation and mitigation measures is urgent. This will counteract environmental degradation and restore ecosystem services to benefit current and future generations.

Finally, it is necessary to reestablish constant monitoring of the MRB with a daily and hourly record by installing a network of automatic hydrometric stations at the outlet of the basin and intersection of its main rivers. Monitoring the MRB will allow for analyzing and understanding the behavior of surface runoff and quality and quantity of water, with which decision-makers will have sufficient information to generate actions aimed at the comprehensive management of the basin's water resources.

**Author Contributions:** Conceptualization, G.C.-G., A.L.-P. and M.A.B.-G.; formal analysis, G.C.-G., A.L.-P. and M.A.B.-G.; methodology, G.C.-G., A.L.-P. and M.A.B.-G.; software, G.C.-G. and A.L.-P.; validation, G.C.-G., A.L.-P. and M.A.B.-G.; writing—original draft, G.C.-G., A.L.-P. and M.A.B.-G.; writing—review & editing, G.C.-G., E.P.-V., A.L.-P, M.A.B.-G., H.F.-M., R.A.-H. and E.I.C.-I. All authors have read and agreed to the published version of the manuscript.

**Funding:** This study was financed by the the Consejo Nacional de Humanidades, Ciencias y Tecnologías (CONAHCYT) for funding the Ph.D. studies of G.C.-G. (Scholarship no. 586381).

**Data Availability Statement:** The data presented in this study are available upon request to the corresponding author.

**Acknowledgments:** The authors wish to thank the Consejo Nacional de Humanidades, Ciencias y Tecnología (CONAHCYT) for the scholarship granted to G.C.-G. for his doctoral studies, as well as to the Hydrosciences Postgraduate Course of the Colegio de Postgraduados for the support provided in the development of this research. Finally, a special gratitude to INIFAP and MSc Walter Lopez-Baez for providing me with the necessary support to successfully complete my investigation on the hydrological modeling in the area of water management resources.

**Conflicts of Interest:** The authors declare that they have no conflict of interest.

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
