# Peer review of "Evaluation of the Impact of Climate Change on the Water Balance of the Mixteco River Basin with the SWAT Model"

_hydrology, doi:10.3390/hydrology11040045_

Round 1

Reviewer 1 Report

Comments and Suggestions for Authors

The paper is very well organised and written, but, from my point of view, no new methodology is applied that can contribute to the scientific community.

I understand that the baseline simulation were made with daily precipitation and the calibration was made with monthly runoff. I think it would be interesting if the authors could explain exactly which precipitation and temperature data were used in SWAT to simulate short-term, mid-term and long-term changes in those 25 years (resolution). Have the authors applied some novel methodology to disaggregate precipitation and/or temperature data? If yes, please explain in the paper. Maybe in the methodology used to disaggregate precipitation and temperature data could be the new contribution. Please, search around and incorporate new contributions to the scientific community in order to consider this paper as a research one.

Reviewer 2 Report

Comments and Suggestions for Authors

Dear Authors,

Title: Evaluation of the impact of climate change on the water balance of the Mixteco river basin with the SWAT model.

This paper covers an interesting subject and contributes to the academic field. Evaluating the impact of climate change on the water balance using different GCMs and SWAT models is highly important. Unfortunately, the article has some limitations and mistakes that must be corrected. The GCMs were used without downscaling, which is critical in such studies. Also, one bias correlation technique was used without discussing different techniques. Which grid/station was selected for statistical metrics and why? Also, the article lacks a detailed discussion of the main results, which is very important in this type of article. For example, why is a specific model the best model? I believe it should be accepted with major revision at this time. It could come much better with a higher impact. I list out some main concerns below, and then the comments for the lines.

Major comments:

The hydrological and meteorological data and modeling are done for the period between 1970 – 1980. Is it sufficient? In such studies, a period of 30 years is preferable. You must discuss the period point within the manuscript.

The resolution of GCMs is relatively high (more than 100 km), so downscaling is a crucial step before doing anything. Revise the article based on this point and make the requested modifications.

 The number of grid points resulting from the GCMs and the resolution of the original data are totally different. How did you select the stations/grids for the bias correlation process?

There are many types of bias correlation. They differ and have many differences. Why did you prefer to use this method without mentioning its pros and cons?

Table 3: each statistical matric was calculated for each grid/point, which can be 20, 100 points within the study area. Which value did you use in this table? And is this value representative?

The selection process of the best GCM in the study area must be discussed in more details.

Minor comments:

Lines 93 – 104: Kindly revise and rewrite it again.

Line 107: is instead of was.

Line 110: write that several GCMs will be used.

Line 151: how did you calculate the Q surface and W seep?

Line 157: What is the resolution of each HRU?

Line 162: Did you calculate the interception as a single value?

Table 1: change and modify the meteorological and hydrometric data with more details. For example, precipitation, temperature, etc. Also, what is the Journal?

Line 178: Modify the square.

Lines 179 – 181 need more explanation.

Line 234: T obs/ P obs, is it an in-situ meteorological station? Or a grid data?

Equations 4-7: You checked and found the metrics for 1970 – 1980 between corrected historical data from GSMs and the observation data. Which grids/points did you use for this step? And is 10 years is sufficient?

Line 250: Which program did you use to create the Tylor program?

The Tyloe results need more explanation. All GCMs are located at the same point.

 Line 331: explain and provide a specific section for the validation and calibration process.

 Line 332: table 4. Modify it.

Table 4: why is the surface runoff low?

Why did you calculate both ET and PET?

Table 5: mention which model you use in the caption.

 Sincerely,

Comments on the Quality of English Language

Minor editing of English language required

Reviewer 3 Report

Comments and Suggestions for Authors

The manuscript may be corrected as suggested.

Round 2

Reviewer 1 Report

Comments and Suggestions for Authors

The paper is very well organised and written, but, from my point of view, no new methodology is applied that can contribute to the scientific community. The conclusion is that, considering the GCM, a high decrease of the water yield will produce (50-60%) and measures are needed.

Only 9 years of monthly precipitation and flow data are used for calibrating and validating, so the adjusted parameters probably have much uncertainty. Another important conclusion could be about the importance of collecting hourly or at least daily flow data.  

It is clearly a very well-made technical report, but I am not considering it as a research paper.

Reviewer 2 Report

Comments and Suggestions for Authors

Dear Authors,

I would like to thank the authors for their efforts and work. They addressed most of my concerns and questions. The revisions made to the manuscript have significantly enhanced its quality and clarity. However, I have specific points that require further clarification or adjustment. I kindly request that you consider addressing these points in your final revisions.

·       Mention in the discussion section that data availability is one of the main limitations of this research. So, the period 1970 – 1980 was used.

·       Line 266: The GCM section must be expanded.

·       The conclusion section is the fourth section, not 5th. Revise the whole article accordingly.

Sincerely,

Comments on the Quality of English Language

Minor editing of English language required
